Under which conditions can introverts achieve happiness? Mediation and moderation effects of the quality of social relationships and emotion regulation ability on happiness

Cabello Rosario 1
Fernandez-Berrocal Pablo 2 pabloberrocal@gmail.com
1 Department of Psychology, Faculty of Education Science, University Castilla-La Mancha , Ciudad Real , Spain
2 Department of Basic Psychology, Faculty of Psychology, University of Malaga , Malaga , Spain
Zohar Ada
Electronic publication date: 2015 Oct 8
Publication date: 2015
Volume: 3
Electronic Location ID: e1300
Received 2015 Jul 3; Accepted 2015 Sep 16
Copyright: © 2015 Cabello and Fernandez-Berrocal
Copyright year: 2015
Copyright holder: Cabello and Fernandez-Berrocal
License: This is an open access article distributed under the terms of the Creative Commons Attribution License, which permits unrestricted use, distribution, reproduction and adaptation in any medium and for any purpose provided that it is properly attributed. For attribution, the original author(s), title, publication source (PeerJ) and either DOI or URL of the article must be cited.
License URL: https://creativecommons.org/licenses/by/4.0/

Keywords: Happiness, Extraversion, Moderate, Mediate, Quality of social relationships, Emotion regulation ability

Funding: SEJ-07325 PSI2012-37490 Rosario Cabello and Pablo Fernández-Berrocal were supported in part by projects SEJ-07325 and PSI2012-37490 (Spain). The funders had no role in study design, data collection and analysis, decision to publish, or preparation of the manuscript.

==============================
Personality traits have been directly associated with happiness. One consistent finding is a strong link between extraversion and happiness: extraverts are happier than introverts. Although happy introverts exist, it is currently unclear under what conditions they can achieve happiness. The present study analyzes, generally, how the quality of social relationships and emotion regulation ability influence happiness and, specifically, how these factors can lead introverts to be happy. In the present study, 1,006 participants aged 18–80 (42% males) completed measures of extraversion, neuroticism, quality of social relationships, emotion regulation ability, and happiness. We found that extraverts had significantly higher happiness, quality of social relationships and emotion regulation ability scores than introverts. In addition, people with high quality social relationships or high emotion regulation ability were happier. Serial mediation analyses indicated that greater levels of extraversion were associated with greater happiness, with small effect size, via two indirect mechanisms: (a) higher quality of social relationships, and (b) higher quality of social relationships followed serially by higher emotion regulation ability. We also found a moderating effect due to the three-way interaction of extraversion, quality of social relationships, and emotion regulation ability: introverts were happier when they had high scores for these two variables, though the effect size was small. These results suggest that the quality of social relationships and emotion regulation ability are relevant to our understanding of complex associations between extraversion and happiness.

Introduction

Personality is one of the personal factors that most influences happiness, as Aristotle (trans. 1984) first pointed out 2,300 years ago and as scientific studies of the 21st century confirm (Diener & Seligman, 2002). Research has indicated a strong influence of extraversion on happiness, showing that extraverts are happier than introverts (DeNeve & Cooper, 1998; Hills & Argyle, 2001; Lucas, 2007; Steel, Schmidt & Shultz, 2008). Although happy introverts exist, it is currently unclear under what conditions they can achieve happiness (Argyle & Lu, 1990). The present study takes a closer look at this question by analyzing how quality of social relationships and emotion regulation ability may positively affect the happiness of introverts, especially in large community-based samples.

Happiness is described as a life involving many pleasant and few unpleasant experiences, and as the experience of high life satisfaction (Schimmack et al., 2004). In general, we apply in the present work a broad conceptualization of happiness including both affective and cognitive components based on Lyubomirsky’s approach (Lyubomirsky, King & Diener, 2005; Lyubomirsky & Lepper, 1999). This perspective attempts to identify people who are global and chronically happy (or unhappy) by directly asking them about their general subjective happiness. This judgment is not merely the simple summation of recent levels of positive and negative affect and satisfaction with life. Although studies have found positive correlations of subjective happiness with life satisfaction and with positive and negative affect, with coefficients ranging from 0.52 to 0.72 (e.g., Lyubomirsky & Lepper, 1999), the correlations are not large enough to justify completely overlaying these concepts.

Although positive affect may seem essential for happiness, and it is often used interchangeably with happiness in the research literature, positive affect does not adequately capture the concept of happiness, it does not guarantee happiness, and it may be unrelated to many of our happiest experiences (Diener & Seligman, 2002; Lyubomirsky, King & Diener, 2005). Nevertheless, several studies have assessed positive affect as an indicator of happiness, and we occasionally draw on those studies in the present work in the absence of studies directly measuring happiness.

We begin our review of the literature by briefly discussing the association between extraversion and happiness, and then discuss how this link may be affected by the quality of social relationships and the ability to regulate emotions adaptively and positively in social interactions.

Extraversion and happiness

Of the five dimensions of personality, extraversion and neuroticism have shown the greatest influence on happiness, with extraversion linked positively and neuroticism negatively to happiness. More specifically, extraversion positively correlates with various happiness measures with correlation coefficients of 0.17–0.45 (Argyle, Martin & Lu, 1995; DeNeve & Cooper, 1998; Cheng & Furnham, 2003). Recently, Cheng & Furnham (2014) showed using a longitudinal design and community sample that extraversion is a significant predictor of happiness, after controlling for parental factors, the subject’s own social factors and intelligence during childhood. Thus extraversion is one of the personality factors most tightly associated with the various cognitive and affective indicators of happiness (Hills & Argyle, 2001; Lucas, 2007; Lucas & Fujita, 2000; Steel, Schmidt & Shultz, 2008). In other words, it is easier for extraverts than introverts to be happy. Nevertheless, happy introverts do exist: Hills & Argyle (2001) found that among 133 participants who reported being happy, 33% were introverts.

Explanations for why extraversion leads to higher happiness indices are controversial, and they range from arguments based on brain structure to arguments grounded in instrumental associations with social relationships (Cremers et al., 2011; Lischetzke & Eid, 2006; Oerlemans & Bakker, 2014; Smillie et al., 2012). Despite the clear association between extraversion and happiness, variables that may mediate and moderate it have yet to be examined systematically and simultaneously, especially in large community-based samples. Two such variables are quality of social relationships and emotion regulation ability.

Quality of social relationships

Studies using instrumental models have shown that social relationships partly explain the relationship between extraversion and happiness (Hills & Argyle, 2001; Argyle & Lu, 1990). Lee, Dean & Jung (2008) found social connectedness (a person’s subjective awareness of being in close relationship with the social world) mediated the association between extraversion and subjective well-being. This is consistent with studies showing that happy people have significantly more full and satisfying interpersonal lives than do people of average happiness or people who are very unhappy (Diener & Seligman, 2002).

Other studies, however, do not support this conclusion and instead report that happiness shows only small correlations with number of friends, frequency of contact, marital status, and actual social activity; effect sizes for these correlations range from 0.10 to 0.20, smaller than those for other variables often interpreted as irrelevant, such as income and health (see Lucas, Dyrenforth & Diener, 2008 for a review). Consistent with the idea that temperament directly mediates the association between extraversion and positive affect, Lucas, Le & Dyrenforth (2008) showed in two studies that even after controlling for differences in levels of social activity and in reactivity to social activity, extraverts still reported greater positive affect than did introverts.

One reason why the existence of social relationships and the amount of time people spend with social relationship partners may not guarantee a high level of happiness is that the social relationships are of insufficient quality. Researchers have proposed that the quality of social relationships can influence the effect of social relationships on happiness (Lyubomirsky, King & Diener, 2005). Hotard et al. (1989) found that strength of social relationships was a strong moderator of subjective well-being in introverted individuals, but not in extraverted ones. More recently, Lucas, Le & Dyrenforth (2008; study 2) analyzed extraversion as a moderator of the association between social activity and positive affect and showed a direct association between extraversion and positive affect. These studies highlight the need to analyze the quality of social relationships as both possible mediator and moderator of the relationship between extraversion and happiness.

Emotion regulation ability

Emotion regulation ability involves being aware of the most effective strategies to modify and nurture emotions in particular situations. It is part of the emotion regulation branch within the Mayer & Salovey (1997) ability model of emotional intelligence. Several reviews describe considerable evidence for the validity of this type of emotional intelligence in different personal and professional contexts (Côté, 2014; Joseph & Newman, 2010). Emotion regulation ability is linked with happiness: people who intelligently regulate their emotions obtain high scores on several measures of happiness (Côté, 2014).

It is unclear whether personality factors moderate the association between the ability model of emotional intelligence and happiness. Several studies failed to find any association between ability emotional intelligence scales and happiness indicators after controlling for personality dimensions (Bastian, Burns & Nettelbeck, 2005; Zeidner & Olnick-Shemesh, 2010). On the other hand, cross-sectional studies (Brackett & Mayer, 2003; Brackett et al., 2006) and prospective studies (Extremera et al., 2011) found that ability emotional intelligence explained 1–4% of variance in happiness measures after controlling for Big Five personality dimensions.

The hypothesis that emotional regulation mediates the effect of extraversion on happiness has been examined in some studies using self-reported measures of emotional intelligence or emotional regulation. Lischetzke & Eid (2006) found that mood maintenance, but not mood repair, mediated the effect of extraversion on pleasant–unpleasant trait mood and pleasant–unpleasant affect change. Other studies suggest possible cultural and genetic bases for emotional regulation as a mediator: self-reported emotional intelligence mediated the effects of extraversion on happiness in a community sample in the UK (Chamorro-Premuzic, Bennett & Furnham, 2007), but not in such a sample in India (Hafen, Singh & Laursen, 2011).

Several studies indirectly suggest that emotion regulation ability may be a mediator or moderator of the relationship between extraversion and happiness. Emotion regulation ability has been linked to the quality of social relationships (Brackett et al., 2006; Lopes, Salovey & Straus, 2003; Lopes et al., 2011; Mayer, Roberts & Barsade, 2008 for a review). One study found that, after controlling for personality differences among people with low levels of self-reported emotional intelligence, the level of perceived social support interacted with emotional intelligence to produce higher positive affect (Gallagher & Vellabrodrick, 2008). In contrast, perceived social support and emotional intelligence did not interact to influence subjective well-being. One study showed that adolescents who scored high on extraversion and had greater self-reported emotional intelligence also scored higher on happiness (Salami, 2011). These mediation and moderation hypotheses need to be explored using a measure of emotion regulation ability.

Motivation for the present study

The above literature review provides evidence of direct relationships between happiness on one hand, and extraversion, social relationships and emotion regulation on the other. In fact, studies have suggested that social relationships or emotion regulation ability help mediate or moderate the relationship between extraversion and happiness. However, the evidence is inconclusive: the strength of moderation varies substantially with experimental conditions, and some studies have shown that controlling for personality dimensions eliminates observed correlations of social relationships and emotion regulation ability with happiness. We are unaware of studies analyzing these two variables acting separately and in concert as potential mediators and moderators of the relationship between extraversion and happiness in a large community sample.

Therefore the present study aimed to develop and test different models that consider quality of social relationships and emotion regulation ability as mediator and/or moderator variables that explain or enhance the effects of extraversion on happiness in an adult community sample (Figs. 1 and 2). Emotion regulation ability was evaluated using the Mayer–Salovey–Caruso Emotional Intelligence Test (MSCEIT). This work aims to help fill in several gaps in the research literature. One is the need to address the research question while avoiding the high risk of bias associated with self-reported measures of emotional intelligence or emotion regulation used in most studies on this topic. Self-reported measures do not directly assess people’s emotional abilities but rather people’s self-reported beliefs about their emotional abilities. Self-reported emotional intelligence measures are highly correlated with established measures of happiness and personality and may therefore contain a great deal of unwanted variance (Côté, 2014; Brackett et al., 2006; Webb et al., 2013). Another gap addressed in the present study is whether and how the quality of social relationships affects happiness; most relevant studies have focused instead on the number or amount of social relationships. A third gap is to examine mediation or moderation in a large community sample with people of diverse ages, since most relevant studies have been performed with small undergraduate samples.

Figure 1 Illustration of the indirect effects model for serial mediation.

In this model, quality of social relationships and emotion regulation ability mediate the relationship between extraversion and happiness.

Figure 2 Illustration of the three-way interaction model.

In this model, quality of social relationships and emotion regulation ability moderate the relationship between extraversion and happiness.

Several hypotheses guided the design and interpretation of the present study. First, we expected that, as indicated by the literature, extravert personality, quality of social relationships and emotion regulation ability would be associated with happiness. Second, we expected that quality of social relationships and emotion regulation ability would be related with happiness even after controlling for the influence of extraversion and neuroticism. We included the personality trait neuroticism as a control variable because emotionally stable individuals generally experience greater happiness (DeNeve & Cooper, 1998). Third, we expected that the quality of social relationships and emotion regulation ability would simultaneously mediate the relationship between extraversion and happiness. Finally, and most importantly for the present study, we expected that quality of social relationships and emotion regulation ability would moderate the relationship between extraversion and happiness such that, in line with previous research, introverts with a higher quality of social relationships and higher emotion regulation ability would score higher on happiness scales.

Materials and Methods

Participants

The sample consisted of 1,006 White/Caucasian Spanish volunteers (42% males) aged 18–80 (mean age = 39.53, SD = 14.96, median age = 45, interquartile range = 22–52).

Instruments

Emotion regulation ability

We assessed emotion regulation ability using the MSCEIT 2.0 (Mayer, Salovey & Caruso, 2002; Spanish version by Extremera, Fernández-Berrocal & Salovey, 2006). MSCEIT uses 141 items to assess the four primary abilities (branches) of the Mayer & Salovey (1997) model of emotional intelligence: perceiving emotions in oneself and others, using emotions to facilitate thought, understanding emotional information, and regulating emotions in oneself and others. In the present study, we used only the emotion regulation branch of the MSCEIT, which assesses intra- and interpersonal emotion regulation abilities in separate tasks totaling 29 items. Respondents rate the effectiveness of different strategies for regulating their own feelings in specified situations and for managing emotionally challenging interpersonal situations. Specifically, the items ask respondents to evaluate the effectiveness of various strategies to generate, maintain, or suppress emotions in the situations described. Raw scores on each task were obtained using a consensus scoring criterion, then transformed into standard scores with a mean of 100 and a standard deviation of 15. Confirmatory factor analysis of responses from 2,112 adults has shown that emotion regulation ability is one of four emotional abilities measured by the MSCEIT (Mayer et al., 2003). The split-half reliability of this subscale was measured to be 0.81 (Mayer et al., 2003), with a test–retest correlation of 0.86 (Brackett & Mayer, 2003).

Happiness

Participants were administered the Subjective Happiness Scale (SHS; Lyubomirsky & Lepper, 1999; Spanish version by Extremera & Fernández-Berrocal, 2014), a widely used 4-item global assessment of happiness. Two items ask respondents to describe themselves using both absolute ratings and ratings relative to peers, while the other two items offer brief descriptions of happy and unhappy individuals and ask respondents about the extent to which each description describes them. Each item is assessed on a Likert scale ranging from 1 (not a very happy person) to 7 (a very happy person); for example, Item 1 is “In general I consider myself…”. The SHS has demonstrated good psychometric properties such as test–retest reliability, discriminant validity and convergent validity across 14 samples (Lyubomirsky & Lepper, 1999) and in samples from different cultures (Extremera & Fernández-Berrocal, 2014).

Extraversion and neuroticism

These personality traits or factors were assessed using the extraversion and neuroticism scales of the Big Five Inventory-44 (BFI-44; John & Srivastava, 1999; Spanish version by Benet-Martínez & John, 1998). The BFI-44 is a 44-item, self-report inventory designed to assess the Big Five factors of personality: Extraversion, Agreeableness, Conscientiousness, Neuroticism and Openness to Experience. In the present study we used the eight items of the extraversion scale and the eight items of the neuroticism scale. These scales, like the others on the BFI-44, have shown substantial internal consistency (extraversion α = 0.88, neuroticism α = 0.84), test–retest reliability and clear factor structure.

Quality of social relationships

The Network of Relationships Inventory (NRI; Furman, 1996; Furman & Buhrmester, 1985; Spanish version by Lopes et al., 2011) consists of 30 items assessing different dimensions of social relationships. In our study, we assessed the quality of social relationships using an abridged, 12-item version of the NRI with only one dimension that focuses on companionship, intimacy, affection, and alliance. The dimension includes items such as “how often do you tell this person everything that you are going through?” Participants used a 9-point Likert scale to evaluate to what extent each of the statements occurs in their relations with friends, i.e., with people who are neither family nor romantic partners. The psychometric properties of this scale have been well demonstrated, with a reported Cronbach’s alpha of 0.88 (Furman, 1996; Lopes et al., 2011).

Procedures

Participants were recruited via posters on the local university campus, at retirement homes and in local newspapers. Participants were accepted into the study to ensure a broad, balanced distribution of gender, age, and socio-economic status. Data were collected over two consecutive years with the help of a team of research assistants. Participants completed paper questionnaires individually in small groups under the supervision of research assistants at different educational centers. They received no financial compensation for participation in the study.

The study was carried out in accordance with the Declaration of Helsinki and ethical guidelines of the American Psychological Association. The study protocol was approved as part of the projects SEJ-07325 and PSI2012-37490 by the Research Ethics Committee of the University of Málaga.

Statistical analysis

Preliminary analyses were carried out to compute descriptive statistics and internal consistency, as well as to detect correlations among extraversion, quality of social relationships, emotion regulation ability and happiness. These analyses were carried out using the SPSS package (version 20.0; IBM, Chicago, IL).

To check whether quality of social relationships and emotion regulation ability are related to happiness even after controlling for the influence of extraversion and neuroticism, we conducted four-step hierarchical regression in which gender and age were entered first (as control variables), followed by extraversion and neuroticism, then quality of social relationships, and finally emotion regulation ability. To examine whether quality of social relationships and emotion regulation ability mediate the relationship between extraversion and happiness, we performed serial mediation analysis using Model 6 in the PROCESS tool (Hayes, 2013). PROCESS is an SPSS macro for mediation, moderation and conditional process modeling. It allows for one independent variable, one dependent variable, and more than one simultaneous mediator variable. To directly test our proposed moderation model (Fig. 2), we used Model 3 in PROCESS to develop and analyze a three-way interaction model (Hayes & Matthes, 2009; Preacher, Rucker & Hayes, 2007). The PROCESS macro automatically determines the centering and interaction terms and provides the point estimate and first- and second-order variance estimates of the conditional indirect effect for a given set of moderator values. We used the Johnson-Neyman computational technique (Hayes, 2013) to identify the values of the moderating variable for which the independent and dependent variables showed a significant association.

Results

Sample characteristics and correlations between key variables

Table 1 shows descriptive statistics for the total sample and for men and women separately, as well as the range of variable values, Cronbach’s α and correlation of main variables with age. Age showed significant negative correlations with neuroticism, quality of social relationships and emotion regulation ability, but no correlation with happiness or extraversion. We repeated the analysis treating age as a categorical variable and obtained similar results.

Table 1 Descriptive statistics for total sample and men and women, range, Cronbach’s α and correlation with age of key variables.

	Total	Men	Women			
Variable	M	SD	Range	M	SD	M	SD	α	Correlation with age	
Age	39.53	14.96	18–80	42.9	14.73	37.07	14.64			
Happiness	5.09	1.05	1–7	5.15	1.03	5.04	1.07	.80	.00	
Extraversion	3.45	.69	1.5–5	3.40	.70	3.48	.68	.78	−.02	
Neuroticism	2.96	.73	1.1–5	2.77	.73	3.10	.70	.76	−.08**	
Quality of social relationships	19.66	4.43	3–27	18.90	4.63	20.20	4.20	.94	−.39**	
Emotion regulation ability	98.48	14.15	65.10–134.90	96.58	15.08	99.86	13.29	.80	−.16**	
Notes.

** p < .01.

N = 1,006.

Table 2 shows correlations among the main variables in our study. As expected, we found positive correlations between happiness on one hand, and extraversion, quality of social relationships, and emotion regulation ability on the other. Extraverts received higher happiness scores than introverts. Neuroticism negatively correlated with happiness.

Table 2 Correlations of key variables.

	Correlations	
Variable	1	2	3	4	
1. Happiness					
2. Extraversion	.35**				
3. Neuroticism	−.42**	−.19**			
4. Quality of social relationships	.26**	.20**	−.06		
5. Emotion regulation ability	.18**	.08**	−.03	.20**	
Notes.

** p < .01

N = 1,006.

Four-step hierarchical regression to check whether quality of social relationships and emotion regulation ability are related to happiness even after controlling for the influence of extraversion and neuroticism generated a significant overall model [F(6, 999) = 71.88, p < .001, with an adjusted R2 = .30]. Neither gender nor age was associated with happiness. The personality traits of extraversion (β = 0.23, p < .001) and neuroticism (β = − 0.35, p < .001) were, respectively, positively and negatively associated with happiness (ΔR2 = .24). In addition, both quality of social relationships (β = 0.20, p < .001; ΔR2 = .04) and emotion regulation ability (β = 0.13, p < .001; ΔR2 = .02) were positively associated with happiness.

Mediation analyses

We created a serial multiple mediation model (Hayes, 2013) using quality of social relationships and emotion regulation ability as mediators. Gender, age, and neuroticism were controlled throughout these analyses. In serial mediation, mediators are assumed to have a direct effect on each other (Hayes, 2013), and the independent variable (extraversion) is assumed to influence mediators in a serial way that ultimately influences the dependent variable (happiness). As illustrated in Fig. 1, a total effect (c) refers to the relationship between extraversion and happiness without controlling for mediators; a direct effect (c′), to the relationship between extraversion and happiness after controlling for mediators; a total indirect effect (ab), to the role of two mediators in the relationship between extraversion and happiness; and a specific indirect effect (a1b1 and/or a2b2), to the role of a particular mediator in the relationship between extraversion and happiness. From our serial multiple mediation model involving quality of social relationships and emotion regulation ability as mediators, we obtained three specific indirect effects through (1) the quality of social relationships (a1b1), (2) the quality of social relationships and emotion regulation ability (a1a3b2), and (3) emotion regulation ability (a2b2) (Fig. 1).

The results showed significant total (c) or direct effects (c′) of extraversion on happiness (Table 3). The total indirect effects of extraversion (ab) were statistically significant, since the 95% confidence interval (CI) of the point estimate did not cross zero. Two significant specific indirect effects were also found. First, there was a significant indirect pathway for extraversion through quality of social relationships (a1b1). Greater extraversion was associated with higher quality of social relationships, which was itself associated with greater happiness. Second, there was a significant indirect pathway for extraversion through quality of social relationships and emotion regulation ability (a1a3b2). Greater extraversion was serially associated with higher quality of social relationships and emotion regulation ability, both of which were associated with greater happiness. No indirect effect of emotion regulation ability was found for the association of extraversion and happiness (a2b2).

Table 3 Serial mediation analysis to identify direct and indirect effects between extraversion and happiness.

				95% CI	
Effect	Path	Coefficient	SE	LL	UL	
Direct effect of E on QSR	a 1	1.0828***	.1848	.7202	1.4454	
Direct effect of E on ERA	a 2	.0036	.0032	−.0028	.0099	
Direct effect of QSR on ERA	a 3	.0022***	.0005	.0011	.0033	
Direct effect of QSR on H	b 1	.0467***	.0071	.0327	.0606	
Direct effect of ERA on H	b 2	1.9333***	.4107	1.1273	2.7393	
Total effect of E on H, without accounting for QSR and ERA	c	.4185***	.0426	.3348	.5021	
Direct effect of E on H when accounting for QSR and ERA	c′	.3564***	.0419	.2743	.4386	
Total indirect effect	ab	.0620	.0149	.0353	.0946	
Indirect via QSR	a 1 b 1	.0505	.0136	.0272	.0810	
Indirect via QSR and ERA	a 1 a 3 b 2	.0046	.0018	.0019	.0096	
Indirect via ERA	a 2 b 2	.0069	.0069	−.0054	.0224	
Happiness total effect modela (R2 = .25***)						
Notes.

Coefficient nonstandardized B coefficients

SE standard errors

CI bias-corrected and accelerated 95% confidence interval

LL lower limit

UL upper limit

E extraversion

H happiness

QSR quality of social relationship

ERA emotion regulation ability; 10,000 bootstrap samples

a Age, sex, and neuroticism were covaried.

*** p < .001.

N = 1,006.

Moderation analyses

We used extraversion, quality of social relationships and emotion regulation ability as indicator variables and happiness as the criterion factor (Fig. 2). Gender, age, and neuroticism were included as control variables. We tested all two-way interactions (fifth-step; extraversion × quality of social relationships, extraversion × emotion regulation ability, quality of social relationships × emotion regulation ability) and the three-way interaction (sixth-step; extraversion × quality of social relationships × emotion regulation ability). Moderation analysis on happiness revealed a significant three-way interaction among extraversion, quality of social relationships and emotion regulation ability (β = − 0.062, p < .01; ΔR2 = .01). Two-way interactions between these variables were not significant.

Figure 3 shows regions of significance for happiness using the Johnson–Neyman technique. Among introverts, defined as those scoring at least 1 SD below the mean on the extraversion scale, quality of social relationships significantly increased self-reported happiness in those who had above-average levels of emotion regulation ability. This variable began to exert a moderating effect at emotion regulation ability scores above 113.15 (t = − 1.96, p < .05). In particular, introverts with a high quality of social relationships and an emotion regulation ability score above 113.15 obtained a mean happiness score of 5.25, which was higher than the mean score of 4.65 for all introverts, and 1 SD above the mean score of 4.28 for introverts with a low quality of social relationships and low emotion regulation ability. As Fig. 3A shows, there was a positive effect of quality of social relationships on happiness for introverts with high emotion regulation ability (b = 0.067, t = 4.07, p < .001). However, there was no effect of quality of social relationships on happiness for introverts with low emotion regulation ability (b = 0.034, t = 1.62, p > .10).

Figure 3 Three-way interaction model to examine how the introversion/extraversion dimension affects happiness.

In this model, the quality of social relationships and emotion regulation ability moderate the relationship between extraversion and happiness. Results are shown separately for introverted (A) and extraverted (B) participants. ∗∗∗p < .001. N = 1,006.

In contrast, among extraverts, defined as those at least 1 SD above the mean on the extraversion scale, quality of social relationships significantly increased reported happiness in all individuals except those with very low emotion regulation ability. This variable began to exert a moderating effect at emotion regulation ability scores below 76.15 (t = 1.96, p < .05). More specifically, extraverts with a low quality of social relationships and emotion regulation ability scores below 76.15 obtained a mean happiness score of 4.80, which was below the mean score of 5.49 for all extraverts, and 1 SD below the mean of 5.85 for extraverts with a high quality of social relationships and high emotion regulation ability. As Fig. 3B shows, there was a positive effect of quality of social relationships on happiness for extraverts with low emotion regulation ability (b = 0.068, t = 4.22, p < .001). However, there was no effect of quality of social relationships on happiness for extraverts with high emotion regulation ability (b = 0.025, t = 1.57, p > .10).

Discussion

The present findings support the existence of several relations between happiness on one hand and extraversion, quality of social relationships, and emotion regulation ability on the other. The pattern of correlations indicates that individuals with more extraverted and less neurotic personalities report higher happiness. This finding confirms previous research reporting a strong association of happiness with the personality factors extraversion and neuroticism (Cheng & Furnham, 2014; DeNeve & Cooper, 1998; Hills & Argyle, 2001; Lucas, 2007; Steel, Schmidt & Shultz, 2008). Even after controlling for these two personality factors, as well as for gender and age, quality of social relationships and emotion regulation ability still showed a significant association with happiness. Thus, our study suggests that people with a high quality of social relationships or high emotion regulation ability are happier, consistent with previous studies (Diener & Seligman, 2002; Brackett et al., 2006).

The presence and small effect size (2–4%) of correlations between happiness and quality of social relationships and emotion regulation ability are consistent with research supporting the relevance of social relationships on happiness (Hotard et al., 1989; Lee, Dean & Jung, 2008), as well as with the reported discriminant and predictive validity of emotional intelligence evaluated as an ability (Brackett & Mayer, 2003; Brackett et al., 2006; Extremera et al., 2011; Mayer, Roberts & Barsade, 2008; Webb et al., 2013). At the same time, our study provides important new insights. Whereas previous work focused on how social relationships or higher emotion regulation ability on their own affect the association between extraversion and happiness, we examine these factors in conjunction and provide the first evidence that they may simultaneously influence happiness.

Our results suggest that extraversion influences happiness through direct and indirect processes. In our serial mediation model, extraversion had a direct effect on happiness, which is consistent with previous research establishing a concurrent association between the personality trait extraversion and happiness (DeNeve & Cooper, 1998). In addition, we found that greater levels of extraversion were associated with greater happiness (a) indirectly via a higher quality of social relationships, and (b) both indirectly and serially via higher quality of social relationships and higher emotion regulation ability. This is consistent with previous research proposing that social relationships mediate the association between extraversion and happiness (Argyle & Lu, 1990; Lee, Dean & Jung, 2008). Those studies focused more on the amount of social relationships, whereas we focused on the quality of relationships. In this sense, our study complements and extends the literature.

Our results with emotion regulation ability are consistent with findings from studies suggesting that emotional regulation or self-reported emotional intelligence mediate the link between extraversion and happiness (Lischetzke & Eid, 2006; Chamorro-Premuzic, Bennett & Furnham, 2007; Hafen, Singh & Laursen, 2011). Those studies relied on self-reported measures of emotional intelligence; we, in contrast, used the MSCEIT and focused on emotional intelligence as an ability measure. In this way, our results significantly extend the literature. Indeed, our analysis is the first to provide evidence that mediation occurs serially via higher quality of social relationships. This may mean that the ability to regulate emotions based on experience strengthens social attachments and avoids friction with friends, contributing to positive and successful social interactions that generate more happiness (English et al., 2012; Lopes, Salovey & Straus, 2003; Lopes et al., 2011).

In addition to providing insights into how quality of social relationships and emotion regulation ability may mediate the association between extraversion and happiness, our study performs the first analysis of three-way interactions among the variables. Our goal was to determine whether quality of social relationships and emotion regulation ability moderate the effects of extraversion on happiness. As expected, we found evidence of weak but significant moderation: introverts were happier when they had high quality of social relationships and high emotion regulation ability. Extraverts were less happy when they had a low quality of social relationships and low emotion regulation ability, but not when they had high emotion regulation ability. These findings suggest that in introverts, high emotion regulation ability reinforces the positive effects of quality of social relationships, thereby increasing happiness. In fact, introverts in our study who had high quality in their social relationships and high emotion regulation ability showed a mean happiness score of 5.25, which is above the mean for the general Spanish population (M = 5.09; Extremera et al., 2011). In extraverts, low emotion regulation ability weakens the positive effects of quality of social relationships on happiness, suggesting that extraverts with low emotion regulation ability may find it difficult to obtain positive experiences from their social relationships. These findings are consistent with previous research proposing that strength of social relationships moderates the association between extraversion and subjective well-being (Hotard et al., 1989), and with other studies concluding that self-reported emotional intelligence moderates the association between extraversion and happiness (Salami, 2011) or between social relationships and life satisfaction (Gallagher & Vellabrodrick, 2008). Therefore, even small changes to social relations or an individual’s abilities can have important effects for the course of his or her life (Brackett & Mayer, 2003; Brackett et al., 2006; Extremera et al., 2011).

Limitations and future directions

The cross-sectional nature of our study and its reliance on a single instrument to assess different constructs precludes inferences about causality in the relationships among variables studied. While our mediation analyses support the notion that quality of relationships and emotion regulation ability lead to greater happiness, prospective studies are needed to address this question rigorously. Nevertheless, our data clearly suggest that researchers should consider alternative mediation and moderation models when analyzing relationships between personality factors and happiness.

Though brief, the SHS measure of happiness is one of the most widely used instruments to assess subjective happiness in different languages around the world (Extremera & Fernández-Berrocal, 2014). Nevertheless, brief self-report measures of happiness have their limitations. Future studies should also explore and replicate these findings using other approaches to measure well-being and, for example, comparing hedonic and eudaimonic views of happiness.

Our data indicate a quite small (1%) effect size for the three-way interaction among extraversion, quality of social relationships and emotion regulation ability. This result should therefore be considered preliminary and verified in future studies, preferably in different ethnicities. Nevertheless, we believe that this 1% interaction has real-world significance because it emerged from sixth-order regression, and because interactions of 1–5% between well-established constructs are considered meaningful by many researchers (Shulman & Hemenover, 2006).

It would be interesting in future research to examine how emotion regulation ability interacts with social relationships and personality dimensions (e.g., extraversion and neuroticism) to influence variables related to mental health, such as anxiety, depression, perceived stress or consumption of addictive substances. For instance, it may be fruitful to examine whether interactions of some of these variables are associated with reduced symptomatology for these pathologies.

Future studies should also explore the effect of cultural dimensions such as individualism/collectivism (Ogihara & Uchida, 2014) on how much the quality of social relationships and emotion regulation ability influence the relationship between extraversion and happiness. Other mechanisms related to emotion regulation ability, such as self-control (Cheung et al., 2014), emotional regulation strategies (Cabello et al., 2013) and implicit theories (Cabello & Fernández-Berrocal, 2015), may also be important for understanding the relationship between extraversion and happiness.

Finally, longitudinal interventional studies should be undertaken to test whether emotional regulation training can help both introverts and extraverts construct better social relations and become happier. Such research may provide more insight into experience- and age-dependent evolution in the association between extraversion and happiness, as well as in the influence of factors like quality of social relationships and emotion regulation ability on that association (Cabello et al., 2014). Evidence suggests that training in social and emotional competencies is crucial and should begin in the first years of life (Durlak et al., 2011).

Conclusions

This exploratory investigation is the first to test different models that consider quality of social relationships and emotion regulation ability evaluated by MSCEIT as mediator and/or moderator variables that enhance the effects of extraversion on happiness in a large adult community sample. Previous authors have noted that the direct, indirect, and interaction effects of several variables affecting happiness are not mutually exclusive. In other words, the association between extraversion and happiness is likely to be multiply determined (Lischetzke & Eid, 2006; Lucas & Baird, 2004). Our research investigated these mediation and interaction processes and revealed that the quality of social relationships and emotion regulation ability are relevant to our understanding of complex associations between extraversion and happiness.

Our study, together with the literature from which it emerged, suggests that many things can strongly influence one’s happiness, including personality traits such as extraversion and neuroticism. Nevertheless, even introverts can achieve happiness if they succeed in developing a high quality of social relationships and in managing their emotions intelligently (Diener & Seligman, 2002; Sheldon & Lyubomirsky, 2007; Mayer, Salovey & Caruso, 2008).

Supplemental Information

Supplemental Information 1 Raw data

Click here for additional data file.

Additional Information and Declarations

Competing Interests

Author Contributions

Human Ethics

The authors declare there are no competing interests.

Rosario Cabello conceived and designed the experiments, performed the experiments, analyzed the data, contributed reagents/materials/analysis tools, wrote the paper, prepared figures and/or tables, reviewed drafts of the paper.

Pablo Fernandez-Berrocal conceived and designed the experiments, analyzed the data, contributed reagents/materials/analysis tools, wrote the paper, reviewed drafts of the paper.

The following information was supplied relating to ethical approvals (i.e., approving body and any reference numbers):

The study was carried out in accordance with the Declaration of Helsinki and ethical guidelines of the American Psychological Association. The study protocol was approved as part of projects SEJ-07325 and PSI2012-37490 by the Research Ethics Committee of the University of Málaga.

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
