# Peer review of "Under which conditions can introverts achieve happiness? Mediation and moderation effects of the quality of social relationships and emotion regulation ability on happiness"

_PeerJ, doi:10.7717/peerj.1300_

## Round 0.1 · original submission · Major Revisions

The study you present in the paper is a large-scale study of personality and happiness in the general population and as such of interest to the reader of personality and of happiness research.

However it has some major weaknesses that need addressing.
You will find the two reviewers comments which I would like you to address and describe your treatment of in your rebuttal letter should you choose to resubmit.

I would also like you to address my criticisms as follows:

1. The title for this paper is intriguing. When are introverts happier? We all know the basic result that extroverts are happier. Unfortunately this paper does not answer this question. It finds, as many other studies find, that extroverts are happier. A less misleading title might be that "under which conditions are introverts less unhappy?" though I acknowledge this title is not half as catchy it is less misleading.

2. In addition the title suggests that you have understood the causal temporal process, thus "when" and not associated states or traits. Since this was a cross-sectional study I find the use of "when" additionally misleading.

As for the research design, it seems you invested much more effort in measuring social support (30 items) than in any of the psychological traits that star in the results (4 items for happiness; 8 for extraversion). There is a big literature on how to think about and how to measure happiness and personality, it seems to me that you should mention in the introduction why you chose this definition and approach to happiness and in the limitation of the discussion include the potential measurement problems.

·

Basic reporting

* The abstract is clear, although I recommend adding that extroversion correlated significantly with quality of social relationships and emotional regulation.
* The introduction is excellent but too long. It presents a clear overall picture including rationale for the study, but gradually and too late. The opening paragraphs should give the reader an understanding of what the study is about, what is to be investigated and why. After this relevant background can be presented. A rationale is presented briefly in lines 62-64 and lines 76-79 but should come much sooner to give the reader motivation to read on. The hypotheses are stated clearly in the last paragraph of the introduction.
* Tables and figures should include N. In figure 3, A and B should be labeled 'introverted' and 'extroverted' for the sake of clarity.

Experimental design

* Why was happiness chosen to investigate and not satisfaction with life?
* Participants and procedures - I would advise moving the second sentence to the beginning, and including spearate sections for 'participants' and 'procedures' (the latter after the instruments). Procedures should be expanded to better explain what was done. For example were participants invited to a specific place at a specific time to complete them? On paper or online? Project numbers are not needed. Last sentence: 'APA ethical guidelines were adhered to' is enough - this was not an 'overall aim'.
* Instruments - MSCEIT: All in all the information should be organized more clearly. For example, begin by explaining more clearly that there are 29 items that include four primary abilities.
* Statistical analyses - 'relationships' (line 240) should be more specific (correlations).
* I would advise dividing Table 1 into two tables: One presenting descriptive statistics on all measures used - mean and SD, range, alpha chronbach. These should be compared for men and women, and correlation with age reported. The second table should present intercorrelations between all the measures, excluding gender and age. It should be as clear as possible - I find Table 1 is not friendly to the eye.

Validity of the findings

* Statistical analyses and results seem solid. However I must stress that I am not an expert on the mediation and moderation analyses presented, so would defer to the judgement of another reviewer who is.
The Discussion is well written.
* Some literature challenges the idea that extroverts tend to be happier than introverts, questioning how we define happiness.The one limitation that in my opinion has not been given enough consideration in the discussion is the fact that happiness, the central variable of this study, was measured using only one instrument composed of 4 items. It is a pity in my view that at least one other measure was not used, and/or related measures such as satisfaction with life, positive affect. In any case this point needs to be discussed.

Additional comments

This study examines factors that mediate and moderate the personality trait of extroversion. It is important because it adds to our knowledge about the conditions needed by introverts, who usually report lower happiness levels than extroverts on existing scales, to report high levels of happiness (high quality of social relationships, high emotional regulation).
Overall the study is innovative and well written, although the manuscript should be tightened up and minor language issues resolved and some mistakes corrected by a professional editor with English mother tongue.
I feel this paper should be published and would be pleased to read a revised version.

·

Basic reporting

REFEREE REPORT on the manuscript titled:
"Why and when are introverts happier?":mediation and moderation effects of the quality of social relationships and emotion regulation ability on happiness,
by: Rosario Cabello, and Pablo Fernández-Berrocal

( Figures and equations are missing in this submission form- .....see added files)


The paper is well written and the presentation is clear.
The statistical analysis included two different models fitted with the same variables on the same data. One is a mediation model, the second is a moderation model . Each model provided different insights into the relationships between happiness and extraversion with emotion regulation ability and quality of social relationships as mediators or moderators , (and age and gender as control variables).

My report has two main parts. The first includes comments that I consider essential for the revision of the paper. The second are only suggestions that are up to the authors to consider whether to include in the revised version. I do not consider them necessary at all.
FIRST PART
There is an error in line 106. Instead of negative effect should be POSITIVE.
Figure 2 corresponding to the moderation model should be corrected. According to the figure the two moderators differ in some way, while in the theoretical fitted MODERATION model, from what I understand both are represented in the same way.
The model:



The relatively small effect size of the moderating/mediating variables ( QSR and ERA) should be mentioned already in the abstract and not only in the discussion. Also, small effect sizes are obtained not only for the three way interaction, but for all the statistically significant contributions of QSR and ERA.
In Table 2 the notation * corresponds to P<.001. I agree that such a small P is adequate for a large sample in order to infer about significance. However, since usually *is used for .10 , ** for .05 and *** for smaller levels , I suggest to use ***, or to comment about the authors' notation in the text, when referring to the table . In particular, since the bootstrap CI is with .95 level.
Include CI's for the other estimated regression coefficients in this Table.
It is unclear how the authors defined the two categories : introverted (A) and extraverted (B). It should be specified.
Was age included in the model as a continuous variable ? Assuming a linear relationship with age as a continuous variable may be inadequate and is unrealistic. It could explain why it was found to be non-significant.
Check adding age as categorical variable. The appropriate categories can be found by examining a plot of residuals of the model fitted without age, as a function of age.
I also suggest to present the median and interquartile range for age, rather than mean and SD.

SECOND PART
The authors fitted two separate models , one in which QSR and ERA were moderators and one where they were mediators. Both could be easily fitted using available software ( PROCESS). The conclusions were then obtained based on the results of each model separately. It would be much more complicated, but adding some more insight , ( and also is more elegant…) to consider a unifying model that includes the two variables QSR and ERA as both mediators and moderators .



Fitting such a model is not technically straight forward .
Mplus software can be used with Structural Equation Modelling.
A simpler way which may also add insight is to use the SPSS option of fitting a model with only one same mediating and moderating variable. This can be done once with QSR, and once with ERA .
I refer the authors to template 74 from:
Model Templates for PROCESS for SPSS and SAS
c⃝
2013 Andrew F. Hayes, http://www.afhayes.com/
,which I copied from that webside.
One can see in the graph that when X is only included as a mediator without the moderating effect, than the direct effect does not show the path presented by . We also see how the mediating effect depends on the joint values of the explanatory variables , M and X.

Experimental design

No comments

Validity of the findings

No comments

---

## Round 0.2 · accepted · Accept

The paper in its current form is clear and well written. You did a good job of revision. In the abstract please correct extroverts instead of extraverts.